# Jujube Fruit Metabolomic Profiles Reveal Cultivar Differences and Function as Cultivar Fingerprints

**DOI:** 10.3390/plants12122313

**Published:** 2023-06-14

**Authors:** Shengrui Yao, Dikshya Sapkota, Jordan A. Hungerford, Roland D. Kersten

**Affiliations:** 1Department of Plant and Environmental Sciences, New Mexico State University, Las Cruces, NM 88003, USA; dikshya@nmsu.edu; 2Sustainable Agriculture Sciences Center, New Mexico State University, Alcalde, NM 87511, USA; 3Department of Medicinal Chemistry, College of Pharmacy, University of Michigan, Ann Arbor, MI 48109, USA; jhunger@med.umich.edu (J.A.H.); rkersten@med.umich.edu (R.D.K.)

**Keywords:** *Ziziphus jujuba* Mill., jujube, fruit metabolomics, cultivars, location effect, fingerprint, sanjoinine A

## Abstract

Jujube is a nutritious fruit, and is high in vitamin C, fiber, phenolics, flavonoids, nucleotides, and organic acids. It is both an important food and a source of traditional medicine. Metabolomics can reveal metabolic differences between *Ziziphus jujuba* fruits from different jujube cultivars and growth sites. In the fall of 2022, mature fresh fruit of eleven cultivars from replicated trials at three sites in New Mexico—Leyendecker, Los Lunas, and Alcalde—were sampled from September to October for an untargeted metabolomics study. The 11 cultivars were Alcalde 1, Dongzao, Jinsi (JS), Jinkuiwang (JKW), Jixin, Kongfucui (KFC), Lang, Li, Maya, Shanxi Li, and Zaocuiwang (ZCW). Based on the LC–MS/MS analysis, there were 1315 compounds detected with amino acids and derivatives (20.15%) and flavonoids (15.44%) as dominant categories. The results reveal that the cultivar was the dominant factor in metabolite profiles, while the location was secondary. A pairwise comparison of cultivar metabolomes revealed that two pairs had fewer differential metabolites (i.e., Li/Shanxi Li and JS/JKW) than all the other pairs, highlighting that pairwise metabolic comparison can be applied for cultivar fingerprinting. Differential metabolite analysis also showed that half of drying cultivars have up-regulated lipid metabolites compared to fresh or multi-purpose fruit cultivars and that specialized metabolites vary significantly between cultivars from 35.3% (Dongzao/ZCW) to 56.7% (Jixin/KFC). An exemplary analyte matching sedative cyclopeptide alkaloid sanjoinine A was only detected in the Jinsi and Jinkuiwang cultivars. Overall, our metabolic analysis of the jujube cultivar’s mature fruits provides the largest resource of jujube fruit metabolomes to date and will inform cultivar selection for nutritional and medicinal research and for fruit metabolic breeding.

## 1. Introduction

Jujube (*Ziziphus jujuba* Mill.), also called Chinese date or red date, originated in China and it produces a nutritious fruit with sizes varying from thumb size to golf ball size depending on the cultivars. Jujube cultivars were first introduced into the U.S. by Frank N. Meyer from 1908 to 1918 [1]. However, there was limited research on jujubes from 1908 to 2010 in the United States (U.S.) [2,3,4,5,6]. The jujube program of New Mexico State University (NMSU) at Alcalde, New Mexico (NM) started in 2010 and imported jujube cultivars directly from China in 2011 [7]. After 13 years of jujube research at NMSU and cultivar trials at three sites in New Mexico, it has proven that jujube grows and produces well in the U.S., especially in the southwestern region [8,9].

Jujube has been described as a super fruit due to its high contents of carbohydrates such as glucose, fructose, and sucrose; vitamins such as vitamin C; phenolics such as 4-hydroxybenzoic acid glycosides; nucleotides such as cyclic adenosine monophosphate (cAMP) and cyclic guanosine monophosphate (cGMP); minerals; flavonoids such as catechin and quercetin glycosides; and amino acids such as arginine, proline, glutamic acids, and asparagine [10,11]. For example, the vitamin C content of the jujube fruit ranges from 200 to 600 mg/100 g fresh weight and its cAMP content was the highest among 200 fruit species tested in 1980 [12,13,14,15]. In addition, jujube is the source of many natural products with medicinal potential and application in herbal medicine. For example, cyclopeptide alkaloids, which are ribosomally synthesized and posttranslationally modified plant peptides [16,17], have been isolated in large chemical diversity from jujube species, including several compounds with medicinal activities such as anxiolytic sanjoinine A and antiviral jubanine H [18,19,20,21,22]. A cationic peptide called Snakin-Z has also been reported to possess antimicrobial properties in the species [23]. Furthermore, jujube is the source of diverse triterpene chemical compounds, such as pomolic acid and jujubogenin-derived saponins [24]. Consequently, jujube consumption is increasing because of the fruit’s excellent nutritional and potential medicinal contents. Due to time and equipment limitation, most studies on jujube metabolomes mainly focused on limited metabolites.

Mass-spectrometry (MS)-based metabolomics is a powerful technique to provide a global detection of metabolites of biological samples [25,26]. MS-based metabolomics has seen a rapid growth in applications in plant sciences due to developments in liquid chromatography (LC), which allows for the separation of analytes in complex plant extracts prior to MS analysis [27], ion sources for the analyses of labile and chemically diverse metabolites, and mass spectrometer hardware, which enables the rapid acquisition of analytes across wide ranges of analyte concentrations [25,28]. Untargeted metabolomic data analysis is also rapidly improving in the accurate identification rate of analytes in complex plant samples due to advances in LCMS data processing [29,30]. For example, tandem mass spectral annotations via spectral database searches [26,31,32,33,34] or de novo metabolite identification [35] are more commonly used for initial analyte classification and identification. Given the increasing qualitative and quantitative characterizations of analytes in plant metabolomic samples, untargeted plant metabolomics is therefore widely used in species characterization, habitat identification, quality evaluation, growth age and harvest season determination, and in the processing method selection of crops and Chinese medicinal herbs [36,37].

Jujube has been studied by untargeted metabolomics in terms of the dry fruit nutrient composition [38], nutrient dynamics during fruit development, maturation of different cultivars [39,40,41,42], types of flavonoids, the amount of bitterness [42,43], vitamin C [44], free amino acids, polyphenols, sugars [45], and the metabolite differences between different leaf stages [46] and between wild and domesticated jujube [47]. NMSU had replicated jujube cultivar trials at three sites across New Mexico [8,9], which provides an opportunity to evaluate different *Ziziphus jujuba* cultivars in terms of fresh fruit metabolomes within the U.S. We have studied several nutritional metabolites from our jujube cultivars [14,15], but an untargeted metabolomic analysis of these jujube cultivars is lacking. The objectives of this study were, therefore, to examine the global metabolite profiles of mature fruits derived from different jujube cultivars at three cultivar trial sites in New Mexico and evaluate the cultivar-specific effects on jujube metabolomes by liquid chromatography–tandem mass spectrometry (LC–MS/MS) analysis. 

## 2. Results

### 2.1. Jujube Cultivar and Fruit Characteristics

We analyzed 11 jujube cultivars in this study, which included fresh-eating cultivars such as Alcalde 1, Dongzao (HebeiDong and Sandia), Maya, Li, Shanxi Li and Zaocuiwang (ZCW); multipurpose cultivars including Kongfucui (KFC), Jinsi (JS), and Jinkuiwang (JKW); and drying cultivars consisting of Lang and Jixin. Fruit size varied based on cultivar, location, and crop load. In general, the cultivars Maya and Jinsi 2 had fruit weighing < 10 g; the cultivars Jinsi 3, KFC, Jixin, JKW, Sandia, and Lang had fruit weights of 10–20 g from small to large; and the cultivars Li, Shanxi Li, Alcalde 1, and ZCW had fruits larger than 20 g. By the maturation date, Alcalde 1 was the earliest to mature, followed by Maya, ZCW, Jinsi 2, Li/Shanxi Li, Lang, Jinsi 3, JKW, Jixin, and Sandia were the last to mature. The harvest order varied by location and the end uses of those cultivars. Fresh-eating cultivars can be harvested early when they are at the half-red/half-creamy stage or even at the creamy stage, whereas drying cultivars should be harvested at the full-red stage. Exemplary cultivar fruit pictures are displayed in Figure 1.

### 2.2. Total Detected Compounds and Their Categories

Analytes in the UPLC–MS/MS data were identified based on a commercial metabolite database [30] and classified in compound categories based on compound identifications. In total, 1315 metabolites were detected in the 62 jujube fruit samples, with amino acids and derivatives (20.15%, 265 metabolites) and flavonoids (15.44%, 203 metabolites) as the two dominant groups. Phenolic acids, alkaloids, terpenoids, and nucleotides and their derivatives accounted for 12.85% (169 metabolites), 9.35% (123 metabolites), 6.54% (86 metabolites), and 5.17% (68 metabolites), respectively (Figure 2). Among the detected metabolites, natural products (secondary metabolites) represented 48.1% (632 metabolites) of the samples. Figure 3 shows representative metabolite structures of different compound categories derived from jujube. 

### 2.3. Location Effect

To test if location has an effect on the jujube metabolic profile, each jujube cultivar sample was grouped first on the heatmap dendrogram based on the cultivars’ locations (Figure 4). (The full heatmap dendrogram categorized by cultivar and metabolite type is attached as Appendix A). In the dendrogram, the metabolic profiles of 62 samples representing 11 cultivars from three different locations mainly clustered based on cultivar, whereas sample clustering by location was the secondary factor. PCA analysis of the 11 jujube cultivars at three locations showed that samples from Leyendecker and Los Lunas and samples from Los Lunas and Alcalde clustered together, respectively. Leyendecker and Alcalde samples showed less overlap in their PCA plot, indicating a location-specific effect on the metabolic profiles of jujube samples between these growth sites (Appendix A). However, the differentiation of metabolic profiles based on the Leyendecker versus Alcalde location effect was the lowest among all cultivar comparisons, except Shanxi Li/Li and Jinsi/JKW (Section 2.5).

### 2.4. Global Cultivar Metabolic Profiling

Next, we set out to test if the cultivars differed in their metabolic profiles. As mentioned above, the metabolic profiles of the 11 cultivars were mainly grouped by cultivar first (Figure 5). KFC, Jixin, Maya, Dong, and ZCW all clustered together in the corresponding dendrogram heatmap (Figure 5). Samples of Li and Shanxi Li grouped together as well as samples of JS and JKW (see Section 2.5). Additionally, Lang and Alcalde 1 were clustered together, except for one Lang sample from LK and two samples of Alcalde 1 from AL, which separated from other Lang and Alcalde 1 samples, respectively. The Lang sample separated from LK had several high abundance analyte bands in the heatmap. Alcalde 1 and Lang from Alcalde were grouped together with the JS/JKW group. JS and JKW samples from LK and LL were also located together but Jinsi and JKW from Alcalde were separated from others, such as the one sample from Lang–LK. A cultivar-specific metabolite heatmap (Figure 6) shows the cultivar-specific relative richness of each metabolite class (the full-scale metabolite by category heatmap is attached in Appendix A) within each cultivar. For example, our heatmap shows that flavonoids were highest in the cultivar KFC and terpenes were highest in Dong and Maya compared to all other cultivars based on this global metabolic profiling analysis (Figure 6). Specialized metabolites were one of the largest compound classes of differential metabolites among the analyzed jujube cultivars (Table 1).

Principal component analysis (PCA) was performed on all cultivar samples to examine the overall metabolic differences between cultivars and the variation between samples of each cultivar. The all-cultivar PCA analysis showed that Li/Shanxi Li and Jinsi/JKW largely overlapped in their PCA eclipses, corresponding to their sample variation (Figure 7). KFC/Maya metabolomes and Li/Shanxi Li metabolomes were in different areas of the PCA plot than the remaining cultivar metabolomes, indicating significant metabolic differences between Li/Shanxi Li, Maya/KFC, and the remaining cultivars in general.

### 2.5. Cultivar-Specific Pairwise Metabolic Comparison

Given a cultivar-dependent differentiation of jujube metabolomes in our data, we next analyzed all cultivars in their metabolic profiles in comparison to the other jujube cultivars in order to (a) characterize cultivars with the highest metabolic similarity and (b) identify major metabolic differences among cultivars. The relative quantification of significant differential metabolites between all cultivars showed that the most up-regulated significant differential metabolites were from Dong, JS, JKW, Alcalde 1, or Li compared to Maya or KFC (Table 1 and Appendix A). In a 2D PCA of all the metabolic profiles and in an individual PCA, the Shanxi Li/Li data and the Jinsi/JKW data, respectively, clustered together (Figure 5, Figure 7 and Figure 8). This observation is supported by the relatively small number of significant differential metabolites between the Shanxi Li/Li (42 metabolites) and the Jinsi/JKW metabolomes (47 metabolites), respectively (Table 1, Appendix A) compared to the average number of significant differential metabolites between all cultivars being 486. The other cultivars are separated in 2D PCA plots and, in general, differentiated between a higher number of significant differential metabolites ranging from 330 (Alcalde 1 versus Lang) to 575 (Dong versus Maya), indicating that a large fraction of all the detected metabolites is differentiated in these cultivars (i.e., 25–44% of all detected jujube metabolites) (Table 1, Figure 8). As mentioned above, a large fraction of differential metabolites in the jujube metabolomes belongs to specialized metabolism. Overall, specialized metabolite fractions range from 37% (JKW/ZCW) to 56.7% (JX/KFC) of differential metabolites in pairwise cultivar comparisons with >100 significant differential metabolites.

All the 11 cultivars were analyzed in differential metabolites via pairwise comparison, and differential metabolites between two cultivars were grouped into metabolic pathways to identify compound classes which had significant metabolic changes between the two cultivars. The resulting metabolic differences in select cultivar pairs are reported in Table 1 and differences in all-cultivar pairs are shown in the Appendix A.

First, we compared select cultivars with the Alcalde 1 cultivar. Alcalde 1, which is also known as Qiyuexian, is a fresh eating cultivar that was selected from existing trees in 2003 by the Northwest A&F University, Shaanxi, China. Dongzao is a traditional fresh eating cultivar with excellent fruit quality, mainly present in the Shandong and Hebei Provinces, and is now the dominant fresh eating cultivar in China. Alcalde 1 showed 246 up-regulated and 229 down-regulated metabolites compared to Dongzao. The up-regulated metabolites belonged mainly to the biosynthesis of stilbenoids, flavones and flavonols, and glutathione and sulfur metabolism. The down-regulated metabolites were from the amino acid biosynthetic pathways (arginine, alanine, aspartate, glutamate, proline, cyanoamino acid, valine, leucine, isoleucine, D-amino acids), fatty acid metabolism (butanoate, propanoate, linoleic acid), glyoxylate and decarboxylate metabolism, porphyrin biosynthesis, and nitrogen metabolism. Jinsi is popular in regions similar to Dongzao in the Hebei and Shandong provinces, and it is famous for its super drying quality. Alcalde 1 had 192 up-regulated and 183 down-regulated metabolites compared to the Jinsi cultivar. The up-regulated metabolites were compounds of arginine and proline metabolism; nucleotide biosynthesis, including the pyrimidine and purine pathways; phenylpropanoid metabolism, including stilbenoid biosynthesis; fatty acid metabolism, including linoleic acid biosynthesis; the biosynthesis of tropane alkaloids; glutathione metabolism; and betalain biosynthesis. Conversely, the down-regulated metabolites included flavonoids and sphingolipids. KFC was originally derived from the Garden of Confucius in Qufu, Shandong Province, China, and has very good fresh eating quality and can also be used for drying due to its shiny skin. In total, 365 metabolites were up-regulated in Alcalde 1 compared to KFC, whereas 118 metabolites were down-regulated. The up-regulated metabolites belonged to phenylpropanoid metabolism, including flavonoids, flavones, flavonol, and stilbenes, whereas down-regulated metabolites included intermediates of tryptophan metabolism.

Lang is one of two dominant commercially available cultivars in the U.S. that was imported in 1908 and is good for drying. Alcalde 1 had 212 up-regulated metabolites and 118 down-regulated metabolites compared to Lang. The up-regulated metabolites contained compounds from phenylpropanoid metabolism including flavonoids; from amino acid metabolism, including the tryptophan and phenylalanine metabolic pathways; and from glutathione biosynthesis. Li is the most popular commercially available cultivar in the U.S., which is good for fresh eating. In this case, Alcalde 1 metabolic profiles showed 206 up-regulated metabolites and 130 down-regulated metabolites. The up-regulated compounds included flavonoids, sphingolipids, and quinones, whereas the down-regulated compounds were nucleotides, isoquinoline alkaloids, and metabolites of the tyrosine pathway. Furthermore, Maya was imported into the U.S. in 2011 by NMSU, and is a popular fresh eating cultivar in the Beijing area, China, with small, football-shaped fruit possessing excellent fruit quality. Finally, Alcalde 1 had 368 up-regulated compounds and 121 down-regulated compounds compared to Maya. In Alcalde 1, metabolites of the phenylalanine pathway and the phenylpropanoid pathway, including flavonoid biosynthesis and arginine and proline biosynthesis were up-regulated, whereas flavones and flavonols were down-regulated.

Next, we compared the cultivar Dong in its metabolic profiles to select cultivars: JKW, KFC, Li, and Maya, respectively. JKW was imported into the U.S. in 2011 by NMSU, and is a 2001 selection from existing Jinsi trees in Canzhou, Hebei Province, China. It is characterized by larger fruit with excellent drying quality. Dongzao showed 316 up-regulated and 248 down-regulated metabolites compared to JKW. The up-regulated metabolites belonged to nucleotide metabolism, including purine and pyrimidine biosynthesis; amino acid metabolism, including arginine, proline, D-amino acid, and phenylalanine biosynthesis; and linoleic acid biosynthesis. The down-regulated metabolites comprised flavones and flavonols. In addition, the Dongzao metabolic profiles included 403 up-regulated and 128 down-regulated metabolites compared to KFC. The up-regulated metabolites included compounds from arginine, proline, phenylalanine and tyrosine biosynthetic pathways, flavonoids, flavones and flavonols, isoquinoline alkaloids, and metabolites of linoleic acid biosynthesis. The down-regulated metabolites were part of glutathione metabolism. Furthermore, Dongzao had 260 up-regulated metabolites and 227 down-regulated metabolites compared to Li. Specifically, amino acids, flavonoids, fatty acids, and sphingolipids were up-regulated, while phenylpropanoids, such as flavones and flavonols, and compounds of sulfur metabolism were down-regulated. Moreover, compared to Maya, Dongzao showed 400 up-regulated compounds and 175 down-regulated metabolites. Linoleic acid, quinones, amino acid metabolites such as tyrosine, arginine, proline, alanine, aspartate, and glutamate, tropane and isoquinoline alkaloids, and nucleotides, such as pyrimidines and purines, were up-regulated in Dongzao.

We further compared JKW to the select cultivars Li, Maya, and ZCW, respectively. JKW had 207 up-regulated metabolites including flavonoids, flavones and flavonols and 245 down-regulated metabolites including fatty acids of the linoleic acid pathway compared to Li. In comparison to Maya, JKW had 392 up-regulated metabolites and 169 down-regulated metabolites. The up-regulated metabolites belonged to the metabolic pathways of phenylalanine, tyrosine, tryptophan, arginine and proline and sphingolipids, while down-regulated metabolites were purines and phenylpropanoids. Additionally, JKW metabolic profiles comprised 160 up-regulated compounds and 224 down-regulated metabolites compared to ZCW. The up-regulated compounds included sphingolipids, flavonoids, flavones and flavonols, whereas down-regulated compounds were metabolites of phenylalanine and purine metabolic pathways. 

Furthermore, we investigated the metabolic differences of the Jinsi fruits to other select cultivars. As mentioned above, Jinsi had a small number of differential metabolites compared to JKW (47 total with 36 up-regulated and 11 down-regulated). In this comparison, the down-regulated metabolites included amino acids, tropane alkaloids, glucosinolates, and 2-oxocarboxylic acids. 

Moreover, Jinsi was analyzed using the differential metabolites in comparison to the JKW, KFC, Lang and Maya cultivars. Compared to KFC, the Jinsi metabolomes had 379 up-regulated metabolites and 115 down-regulated metabolites. Sphingolipids, flavonoids, flavones and flavonols were among up-regulated metabolites, whereas down-regulated metabolites were cyanoamino acids, thiamine, amino acids, nucleotides, coenzyme A metabolites, and purines. In comparison to Lang, 270 metabolites were up-regulated, and 150 metabolites were down-regulated in Jinsi. Compounds from fructose and mannose metabolism as well as nucleotides were up-regulated, whereas linoleic acid, D-amino acids, and phenylpropanoids were down-regulated. Jinsi metabolomes included 394 up-regulated and 139 down-regulated metabolites compared to Maya metabolomes. The up-regulated metabolites were part of amino acid pathways, such as phenylalanine, tyrosine, arginine, and proline. The down-regulated metabolites were nucleotides, purines, and glutathione metabolites.

Jixin was imported into the U.S. in 2011 by NMSU, and is a traditional drying cultivar found mainly in Henan Province, China. It holds a medium-sized fruit with excellent drying quality and shiny skin. Jixin was compared to the KFC cultivar. Herein, 377 metabolites were up-regulated, and 154 metabolites were down-regulated in Jixin. Phenylpropanoid metabolites, such as flavones, flavonols and flavonoids, tropane/piperidine/pyridine alkaloids, and phenylalanine were up-regulated. The down-regulated metabolites were derived from the tryptophan and purine metabolic pathways. 

Moving on, KFC was compared to the U.S. commercial jujube cultivars Lang and Li and to the cultivar Maya, which was imported in 2011. Compared to Lang, KFC metabolic profiles included 178 up-regulated metabolites and 320 down-regulated metabolites. The up-regulated metabolites included glutathione metabolites, whereas down-regulated compounds contained phenylpropanoids such as flavones and flavonols and tyrosine metabolites. In comparison to Li, KFC had 154 up-regulated metabolites and 338 down-regulated metabolites. Herein, glutathione and D-amino acid metabolites were up-regulated in KFC, whereas phenylpropanoids, such as flavonoids, flavones and flavonols, and tropane/piperidine/pyridine alkaloids were down-regulated. Finally, KFC had 222 up-regulated compounds and 252 down-regulated compounds compared to the Maya metabolic profiles. Among the up-regulated metabolic pathways were linoleic acid biosynthesis and tryptophan metabolism. The down-regulated metabolic pathways included biosynthesis of purines, nucleotides, flavonoids, flavones and flavonols. 

Next, we examined the differences between Lang and the select cultivars Li, Maya, and ZCW, respectively. In comparison to Li, Lang had 149 up-regulated metabolites and 206 down-regulated metabolites. The up-regulated metabolites were purines and nucleotides, while the down-regulated metabolites included phenylpropanoids such as flavonoids, isoquinoline alkaloids and metabolites of the phenylalanine and tyrosine pathways. Moreover, compared to Maya, Lang showed 265 up-regulated metabolites and 172 down-regulated metabolites. Many of the up-regulated metabolites were tyrosine, arginine and proline metabolites, phenylpropanoids, and fatty acids, such as linoleic acid, whereas down-regulated metabolites contained many phenylalanine compounds, glutathione, purines, flavonoids, and nucleotides. Finally, Lang contained 181 up-regulated compounds and 264 down-regulated compounds compared to ZCW. The up-regulated metabolites belonged to arginine, proline, glutathione, and tyrosine metabolism, and to the biosynthesis of tropane/piperidine/pyridine alkaloid and phenylpropanoids. The down-regulated metabolites were nucleotides, purines, D-amino acids, and flavonoids.

In addition, the Li fruits were analyzed based on their differential metabolites compared to the select cultivars Maya, Shanxi Li, and ZCW. In comparison to Maya, the Li metabolic profiles included 337 up-regulated metabolites and 149 down-regulated metabolites. Among the up-regulated metabolites were linoleic acids, tryptophan metabolites, and phenylpropanoids. Conversely, among the down-regulated metabolites were purines, glutathione, and nucleotides. As reported above, the metabolic profiles of Li and Shanxi Li revealed a small number of differential metabolites between both cultivars. In total, 36 metabolites were up-regulated and six metabolites were down-regulated. Compared to ZCW, Li showed 198 up-regulated metabolites and 238 down-regulated metabolites. In this case, the down-regulated metabolites included glucosinolates, amino acids, and organic acids, such as 2-oxocarboxylic acids.

The final selected comparison was Maya versus ZCW cultivars. Here, Maya metabolic profiles had 130 up-regulated metabolites, including tropane/piperidine/pyridine alkaloids, and 389 down-regulated metabolites, including phenylalanine, tryptophan, arginine and proline metabolites and flavonoids.

Overall, the pairwise metabolic comparison analysis indicates that the tested jujube cultivars Li (Shanxi Li), Jinsi (JKW), Lang, Alcalde 1, Dongzao, Maya, Jixin, ZCW, and KFC show significantly different metabolic profiles of their mature fruits. The metabolites from the following pathways were most often significantly differentially regulated in the 55 pairwise cultivar comparisons: flavonoid biosynthesis (33/55), linoleic acid biosynthesis (27/55), nucleotide biosynthesis (26/55), purine biosynthesis (25/55), flavone and flavonol biosynthesis (23/55), pyrimidine biosynthesis (20/55), glutathione metabolism (19/55), tropane/pyrimidine/pyridine alkaloid biosynthesis (18/55), tyrosine biosynthesis (12/55), sphingolipid biosynthesis (12/55), and tryptophan biosynthesis (11/55) (Appendix A). A differential metabolic trend regarding the fruit types of analyzed cultivars was the up-regulation of lipid and/or fatty acid biosynthesis in half (9/18) of the drying cultivars compared to the multi-purpose fruit cultivars or fresh fruit cultivars but no down-regulation of lipid and/or fatty acid biosynthesis in any drying cultivars compared to multi-purpose or fresh fruit cultivars (Table 1 and Appendix A).

As mentioned above and also apparent from the pairwise metabolic comparison (Table 1), a large diversity of specialized metabolites was detected in jujube fruits, including medicinal natural products. An example of a medicinal natural product class from *Ziziphus* plants is cyclopeptide alkaloids, a structurally diverse group of cyclic peptides biosynthesized through the ribosomal pathway by copper-dependent BURP peptide cyclases [16] and from predicted split precursor peptides co-localized on the *Ziziphus* genome to hypothetical peptide cyclase genes [17,47]. In the analyzed mature fruit metabolomes, we identified an analyte matching MS and tandem MS data literature values corresponding to sanjoinine A [48], a sedative cyclopeptide alkaloid from *Ziziphus spinosa* which has been characterized as a bioactive compound from jujube herbal medicine used to treat insomnia (Figure 9, Appendix A). Our analysis showed that the putative sanjoinine A analyte was only detected in higher abundance in Jinsi and JKW cultivars, in low abundance in Li and KFC cultivars, and it was not detectable in all other cultivars.

There were several notable detected metabolites from compound classes characteristic for jujube fruits (Figure 3 and Appendix A). Among the 82 detected triterpenes in all jujube metabolomes, pomolic acid and jujubogenin had the highest total ion abundance values in all jujube samples. Both pomolic acid and jujubogenin had the highest relative abundance in the KFC fruits, followed by Dongzao, Alcalde 1, Jixin, and Maya. In addition, among the 68 identified nucleotides and derivatives, the nucleotides with the highest total ion abundance values in all jujube samples were cyclic 3′,5′-guanosine monophosphate (cGMP) and cyclic 3′,5′-adenosine monophosphate (cAMP). Relative cGMP concentrations were highest in Lang, KFC, and Alcalde 1, and cAMP concentrations were highest in Lang and Alcalde 1. Moreover, 79 organic acids were detected in all jujube samples, including malic acid and citric acid having the highest total ion abundance in all samples. Ascorbic acid had the highest relative concentration in KFC. Overall, 169 phenolic acids were characterized in all jujube samples. 4-O-glucosyl-4-hydroxybenzoic acid and salicyloyl-β-D-glucose were the most abundant phenolic acids in all samples with the highest relative concentrations in ZCW, Maya, and Lang. In addition, coniferin was detected in high relative abundance in Jixin and JKW, whereas Jinsi 2, JKW, Lang, KFC, Maya, and ZCW were also rich in trihydroxycinnamoylquinic acid. Finally, 123 alkaloids were detected in the jujube samples, with corydalmine at the highest total ion abundance in all samples (Appendix A). Additionally, catechin was the dominant flavonoid in most cultivars, but KFC and Maya had more 6-hydroxykaempferol-7-O-glucoside and quercetin-3-O-neohesperidoside than catechin; Jinsi 2 also had more quercetin-3-O-neohesperidoside than catechin.

## 3. Discussion

### 3.1. Cultivar Defines Metabolomics Profile and Location Plays a Minor Role

This is the first jujube metabolomics study with more than 10 cultivars from replicated cultivar trials at three locations. There were more metabolites detected (1315) than previous reports with fresh fruit (406) or dry fruit (463) samples [38,39]. The difference observed could be due to different detection methods used or differences in samples, such as the fruit types. In addition, fresh and mature fruits were used in this study while dry fruits were used in others [38]. The increased number of detected analytes indicates a higher metabolic diversity among jujube cultivars than previously reported. A reason for the expanded jujube metabolome in our study could be the improved identification of specialized metabolites in tandem mass spectrometry data aided by improved spectral annotations via growing tandem MS spectral databases [26,31,32,33,34]. The annotation of ~48% of analytes from specialized metabolic categories in all of our jujube metabolomes shows that jujube fruits are rich in diverse natural products.

The southwestern Unites States is an ideal location for jujube production with semi-arid climates, plenty of sunshine, hot summers, and large temperature differences between day and night, which are all similar to the climate conditions in the dominant jujube-producing area—the Xinjiang region in China [9,38]. Shi et al. (2022) reported the nutrient composition and quality of 20 dried jujube fruit samples from several producing areas [38]. They classified the regions as east producing and west producing areas and mentioned five top-quality cultivars, but their cultivar metabolic profiles varied significantly by location. However, our results revealed that samples from the same cultivar cluster together first, whereas a small location effect is only seen for samples between Leyendecker and Alcalde. Thus, each jujube cultivar grown at the New Mexico growth sites, consisting of Leyendecker, Alcalde and Los Lunas, has its unique metabolic profile, which is primarily dictated by the cultivar and then by location.

### 3.2. Cultivar Metabolomic Profile Functions as Cultivar Metabolomic Fingerprint

Our results indicate jujube cultivars vary significantly in their metabolomic profiles, including secondary metabolites, which constitute one of the major groups of differential metabolites among all pairwise cultivar comparisons in our study. Among the eleven cultivars tested, there were two groups with identical DNA single nucleotide polymorphism (SNP) sequences: Li/Shanxi Li and Jinsi (2/3)/JKW [49]. The metabolic similarity of Li/Shanxi Li and Jinsi (2/3)/JKW in their fruit metabolomes agrees with the DNA genotyping results of both cultivar pairs. Therefore, mass spectrometry-based metabolomics enables cultivar differentiation by metabolic fingerprinting. The metabolomic data further demonstrated that Li and Shanxi Li could be synonyms. Li was imported into the U.S. in 1914 by Frank N. Meyer from Shanxi Province, China [7], whereas Shanxi Li was imported roughly 80 years later by Roger Meyer. Shanxi Li is also called Linyi Li because the traditional producing area of Shanxi Li is in Linyi County, Shanxi Province. So, Li, Linyi Li, and Shanxi Li are synonyms of one cultivar or mutations. Similarly, JS and JKW also had identical SNP sequences in a jujube genotyping study [49]. The similarity of their metabolic profiles support their similar genotypes, and both cultivars are closely related cultivars or the same cultivar. Finally, the similar metabolic profiles of Jinsi and JKW cultivars highlight metabolomics as a strategy for cultivar-specific fingerprinting. 

### 3.3. Important Nutritional and Medicinal Components in Jujube Fruits

Our study provides a global metabolomic resource for jujube mature fruit metabolomes, including dry, fresh, and multi-purpose fruits, which can guide the selection of our target cultivars based on the nutritional and medicinal aspects. For example, Alcalde 1, Lang, and KFC, but not Maya and ZCW, should be considered for mature fruits containing high contents of cAMP and cGMP. Moreover, for high ascorbic acid content, KFC is also recommended. The characterization of up-regulated lipid and fatty acid contents in many drying cultivars indicates that this is a metabolic trait that favors drying cultivar production. In addition to nutrients, jujube is a rich source of natural products, which is highlighted by the 632 specialized metabolites identified in all jujube samples in this study. Metabolic profiling can also inform the selection of jujube cultivars for fruits enriched with natural products with medicinal bioactivity. In addition to the sedative sanjoinine A in jujube herbal medicines, Jiang et al. [50] also reported sedative and hypnotic effects caused by saponins and flavonoids extracted from Semen *Ziziphus jujube.*

Our metabolic profiling readily identified two cultivars, Jinsi and JKW, with relatively high content of a sanjoinine A-type cyclopeptide alkaloid among the screened cultivars. Therefore, Jinsi and JKW are the best candidates for sanjoinine A isolation for medicinal research and production and breeding of jujube with this sedative cyclopeptide alkaloid. The presence of the sanjoinine A-type cyclopeptide alkaloid in both JKW and Jinsi is most likely due to their similar genotypes, as mentioned above. Compared to the other tested cultivars, JKW and Jinsi might have precursor peptides with the sanjoinine-type core peptide, which are either lacking in the genomes or the fruit transcriptomes of the other cultivars, resulting in the absence of sanjoinine-type peptides in their fruit metabolic profiles. In addition, the corresponding sanjoinine cyclase or other posttranslationally modifying enzymes of sanjoinine A biosynthesis might not be expressed during the analyzed fruit ripening stage in the other cultivars. Finally, the biosynthesis of sanjoinine A-type cyclopeptide alkaloids in Jinsi/JKW could be a cultivar-specific metabolic trait for abiotic or biotic stress protection of either the fruit or seed.

In conclusion, we determined the major metabolic differences among 11 jujube cultivars grown at three growth sites in New Mexico, United States. Our study showed that the untargeted metabolomic analysis of mature jujube fruits revealed metabolic profiles as cultivar-specific fingerprints, which can inform the selection of cultivars based on nutrient and natural product contents for agricultural and medicinal purposes.

## 4. Materials and Methods

### 4.1. Jujube Sampling

During the jujube fruit harvest season in 2022, samples of 11 cultivars with 2 replications per site and 30 fruits per tree were collected from the existing replicated jujube cultivar trials at three sites in New Mexico as fruit matured from early September to early October 2022 (Table 2) [8,9]. The 11 cultivars were Alcalde 1, Dongzao, Jinsi (Jinsi2/Jinsi3), Jixin, Jinkuiwang (JKW), Kongfucui (KFC), Lang, Li, Maya, Shanxi Li, and Zaocuiwang (ZCW). Cultivars and sample number per site are listed in Table 2. The details for each site are as follows: Alcalde Center—lat. 36°05′27.94″ N, long. 106°03′24.56″ W, elevation 1730 m, USDA hardiness Zone 6a; Los Lunas Center—lat. 34°46′04.7″ N, long. 106°45′45.7″ W, elevation 1478 m, USDA hardiness Zone 7a; and Leyendecker Center (Las Cruces)—lat. 32°12′08.9″ N, long. 106°44′41.4″ W, elevation 1176 m, USDA hardiness Zone 8a [9]. The historic average annual precipitations and average annual temperatures for Alcalde, Las Cruces, and Los Lunas were 251, 234 and 231 mm, and 10.6, 16.5, and 13.0 °C, respectively [51]. The samples were transported back on ice to the lab. Following this, the total weight and soluble solids contents were measured. Around 10 g of fruit wedges from 5 to 6 mature fruits were ground to powder with liquid nitrogen and stored at −80 °C in an ultralow temperature freezer. Samples were lyophilized at the University of Michigan before being sent for metabolomics analysis with the Labcono 4.5 L −84 °C lyophilizer for 48 h. 

### 4.2. Metabolomics Sample Preparation

The sample preparation, extraction, and metabolomics analysis were performed by Metware Biotechnology Inc. (Woburn, MA, USA) based on their standard procedures as previously described [31,51,52]. The samples were re-lyophilized in a vacuum lyophilizer before analysis (Scientz-100F). After that, samples were ground to powder with a ball-mill grinder (MM400, Retsch, Verder Scientific, Inc., Newtown, PA, USA). Subsequently, 1.2 mL of −20 °C pre-cooled 70% methanolic solution was added to 50 mg of sample. Samples were vortexed six times for 30 s every 30 min. Following this, samples were centrifuged at 12,000 rpm for 3 min and the supernatant was filtered through a 0.22 μm membrane and stored at 4 °C for further analysis. In addition, 100 mg of each sample were ground at the University of Michigan with a MP Biomedicals 5×FastPrep Tissuelyzer for 2 min at 6 m/s with 0.2 μm silica beads in 2 mL tissuelyzer plastic screw cap tubes. The ground jujube samples were resuspended in 1 mL 80% methanol, ground for an additional 40 s in the Tissuelyzer at the same speed as before, extracted for 10 min in a 60 °C water bath, centrifuged for 5 min at 16,000× *g*, and finally filtered through a 0.2 μm syringeless PTFE membrane filter (Whatman, Sigma-aldrich, St. Louis, MO, USA). Filtered jujube extracts were either analyzed immediately by LC–MS/MS or stored at −80 °C before analysis.

For the location effect analysis, all the cultivar samples from one location were treated as one group. For the cultivar effect analysis, all the cultivar samples from all three locations were treated as one group, given the small location-specific effects on metabolic profiles identified in the location effect analysis. Jinsi 2 and Jinsi 3 were grouped into one cultivar group (Jinsi).

### 4.3. UPLC–MS Analysis

Ultra-performance liquid chromatography (UPLC) (ExionLC^TM^ AD https://sciex.com/ (accessed on 15 November 2022)) and tandem mass spectrometry (MS/MS) (Applied Biosystems QTRAP 6500, https://sciex.com (accessed on 15 November 2022) were utilized to analyze the jujube fruit samples at Metware Biotechnology Inc, and they were operated under standard procedures, as described in Lozada et al. (2023) [53]. The metabolite extracts were processed by UPLC–MS coupled with a linear ion trap (LIT) and a triple quadrupole-linear ion trap mass spectrometer (Q trap^®^) (AB6500 Q TRAP^®^ UPLC/MS/MS system) with an ESI Turbo ion spray interface. Both cation and anion modes were controlled using the Analyst 1.6.3 software (AB Sciex, Framingham, MA, USA). The operation conditions were as previously described [31,51,52]. 

In addition, all jujube extracts were subjected to LC–MS/MS analysis on a Thermo QExactive orbitrap mass spectrometer coupled to a Vanquish UPLC instrument with the following parameters: injection volume = 2.5 µL; LC—Phenomenex Kinetex 2.6 μm C18 reverse phase 100 Å 150 × 3 mm LC column; LC gradient—solvent A, 0.1% formic acid and solvent B, acetonitrile (0.1% formic acid); cycle conditions—0 min, 10% B; 5 min, 60% B; 5.1 min, 95% B; 6 min, 95% B; 6.1 min, 10% B; 9.9 min, 10% B; 0.5 mL min^−1^; MS—positive ion mode; full MS—resolution of 35,000; mass range 400–1200 m/z; dd-MS2 (data-dependent MS/MS)—resolution of 17,500; AGC target = 1 × 10^5^; loop count of 5; isolation width = 1.0 m/z; collision energy = 25 eV; and dynamic exclusion = 0.5 s.

### 4.4. Qualitative and Quantitative Analysis of Metabolite

A commercial database (Metware’s metabolite database, Metware Biotechnology Inc., Woburn, MA, USA) was used for identification of metabolites [31]. Semi-quantitative analyses of the samples were performed by MRM using QqQ MS. Peak integration on the mass spectrum peaks of all the metabolites identified was performed after obtaining spectrum analysis data from different samples. Multiple injections of the quality control (QC) sample (mixture of a small quantity from each sample) were used in this study to examine the reproducibility of the samples under the same treatment conditions. 

### 4.5. Data Analysis

Principal component analysis (PCA), hierarchical clustering analysis (HCA), and orthogonal partial least squares discriminant analysis (OPLA-DA) are commonly used methods for metabolomics data analysis [54,55]. HCA is a type of multivariate analysis which aims to group/cluster samples with similar characteristics. The heatmaps were generated using the R software heatmap package [56]. The PCA analysis is used to determine whether samples come from different treatment groups or between samples within the same group. HCA and PCA can detect the main effects, but they are not sensitive to variables with small correlations. The orthogonal partial least squares discriminant analysis (OPLS-DA) was implemented to show the differences between each group. HCA, PCA, and OPLS-DA were conducted in R as previously reported [53,55,57,58]. Based on the OPLS-DA analysis results, the variable importance in project (VIP) values were generated for each metabolite. Metabolites with log2FC (fold change) ≥ 1, VIP ≥ 1, and *p* < 0.05 were set as the threshold of significance. The Kyoto Encyclopedia of Genes and Genomes (KEEG) database was explored to link the differential metabolites to metabolic pathways [59]. Metabolites with *p* < 0.05 were considered as significantly different. In addition, the Plant Widely Targeted Metabolome tool from MetwareCloud “Metabonomics_5.0” (Released 31 January 2023) was used for data analysis (https://cloud.metwarebio.com) (accessed from 4 February to 31 May 2023). For differential pathway identification (Table 1), a metabolic pathway was characterized as up- or down-regulated if > 4 metabolites within the pathway were up- or down-regulated, respectively.

Sanjoinine A analysis was conducted with QualBrowser in the Thermo Xcalibur software package (v. 4.3.73.11, Thermo Scientific, Waltham, MA, USA) on sample data obtained at the University of Michigan. The quantitative data analysis and data visualization for sanjoinine A and other metabolites were performed in GraphPad Prism (v9.5.1).

## Figures and Tables

**Figure 1 plants-12-02313-f001:**
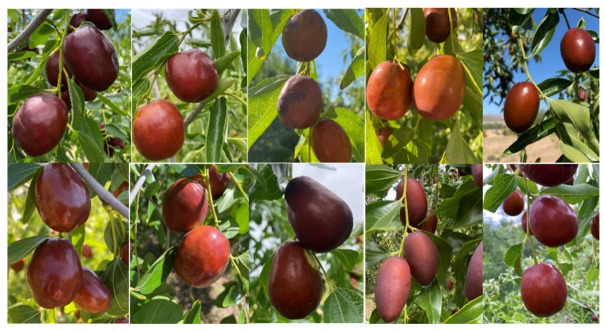
Cultivar fruit pictures. Top row, from left to right: Alcalde 1, Dongzao, Jinsi 3, JKW, and Jixin; bottom row, from left to right: KFC, Li, Lang, Maya, and ZCW. (Pictures taken by S.Y.).

**Figure 2 plants-12-02313-f002:**
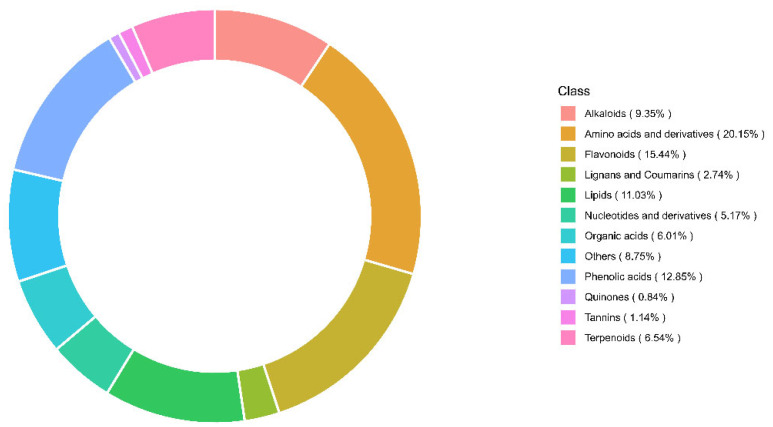
Relative abundance of metabolite classes among 1315 compounds detected in fresh mature jujube samples from New Mexico, United States in 2022.

**Figure 3 plants-12-02313-f003:**
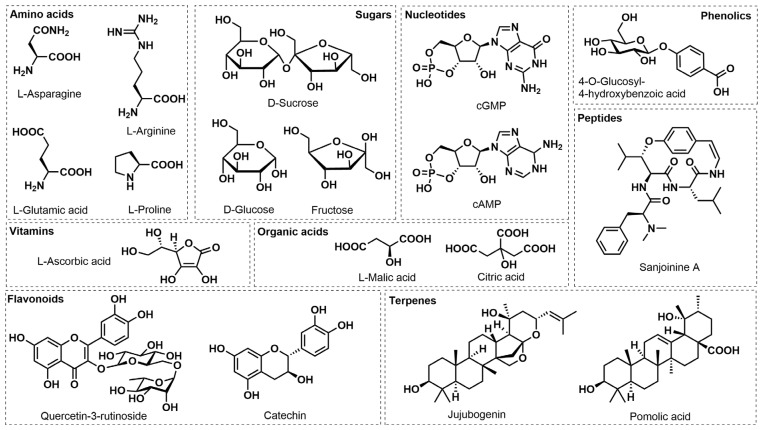
Representative metabolite structures of different categories of compounds found in mature jujube fruit.

**Figure 4 plants-12-02313-f004:**
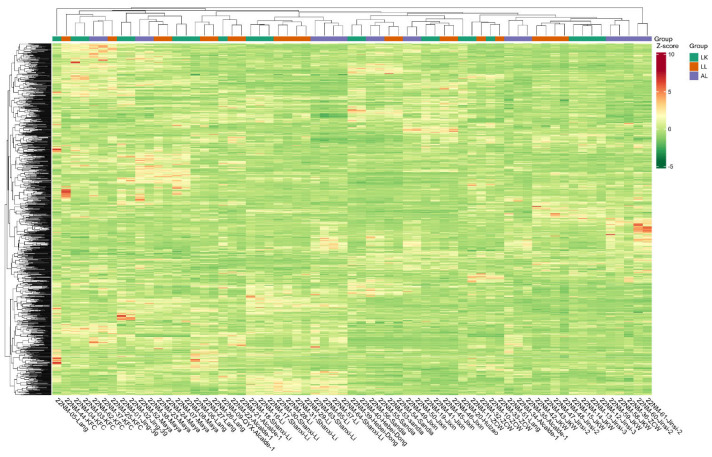
Heatmap containing all cultivars and metabolites with clustering for both locations (top) and metabolites (left). The different colors represent the results after standardization of the relative contents (red represents high content and green represents low content).

**Figure 5 plants-12-02313-f005:**
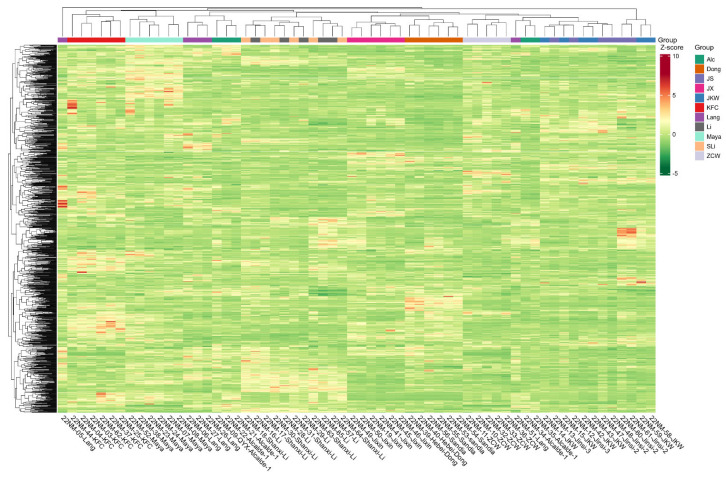
Heatmap of all detected metabolites among all cultivars and clustering analysis for both cultivars and metabolites. The different colors are the results after standardization of the relative contents (red represents high content and green represents low content). The dendrogram on the left side of the figure is the metabolite clustering, and the one on the top of the figure is for cultivar samples.

**Figure 6 plants-12-02313-f006:**
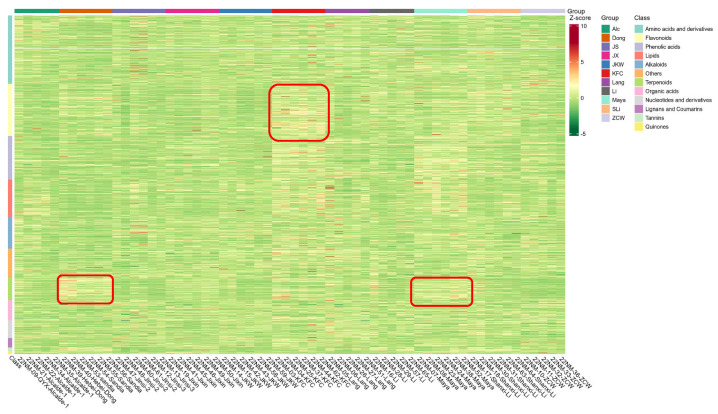
Heatmap for all cultivars and all metabolite classes. The X axis indicates cultivar name and Y axis indicates metabolite classes. The different colors are the results after standardization of the relative contents (red represents high content and green represents low content). Each color bar on the top line represents a cultivar and each color bar on the vertical line represents a metabolite category. The red boxes highlight the high flavonoids in KFC and high terpenes in Dong and Maya.

**Figure 7 plants-12-02313-f007:**
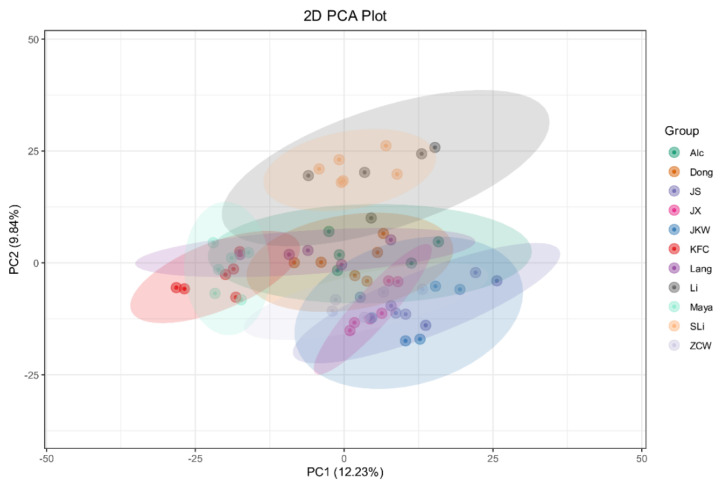
Cultivar principal component analysis (PCA) eclipse figure. Each color represents a cultivar, and each dot represents a sample. PC1 denotes the first principal component and PC2 denotes the second principal component. Percentages represent the interpretation rates applied to the principal components of the data set.

**Figure 8 plants-12-02313-f008:**
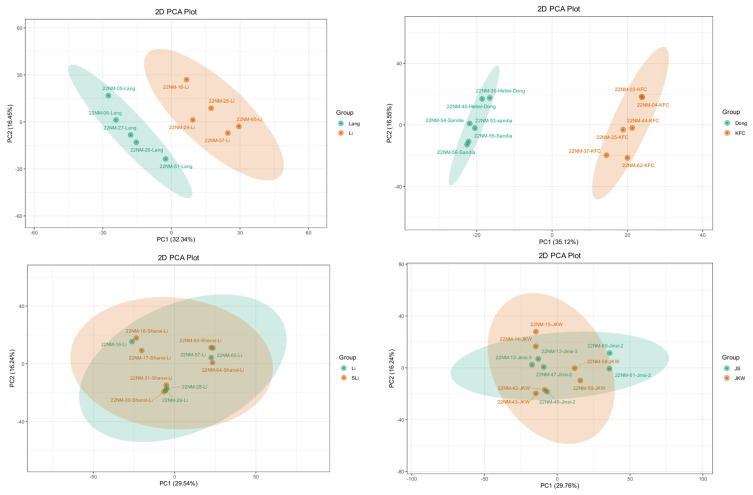
Principal component analysis eclipse of select cultivar comparison. Top row: Li vs. Lang (**left**), Dongzao vs. KFC (**right**). Bottom row: Li vs. Shanxi Li (**left**), Jinsi vs. JKW (**right**). Each color represents a cultivar, and each dot represents a sample. PC1 denotes the first principal component and PC2 denotes the second principal component. Percentages represent the interpretation rates applied to the principal components of the data set.

**Figure 9 plants-12-02313-f009:**
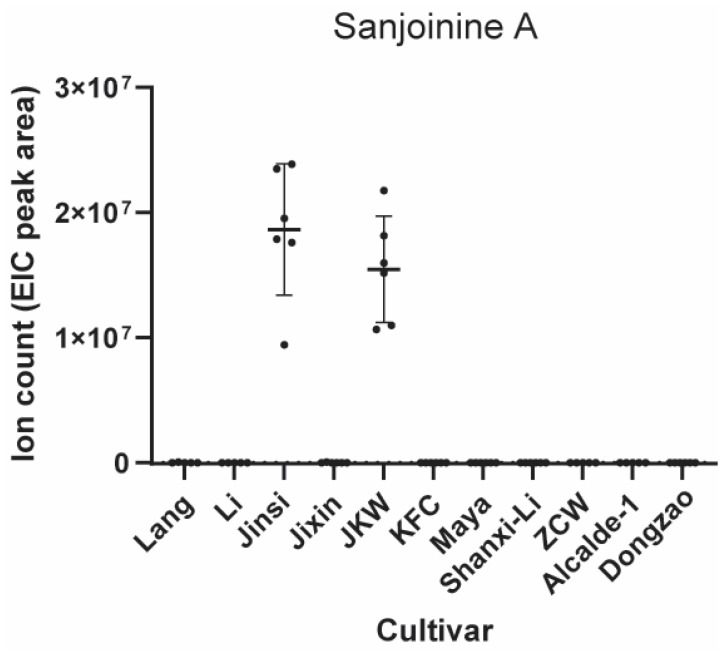
Metabolomic analysis of jujube cultivars for cyclopeptide alkaloid sanjoinine A analyte. Horizontal lines in dot plot represent mean (*n* = 6) and error bars show one standard deviation.

**Table 1 plants-12-02313-t001:** Significant differential metabolites and corresponding metabolic pathways between cultivars. If A vs. B is down-regulated, A has more metabolite than B; if up-regulated, B has more metabolite than A. Abbreviations: MP—multipurpose.

Cultivar Groups (Fruit Types)	All Significant Differential Metabolites	Up-Regulated Metabolites	Down-Regulated Metabolites	Up-Regulated Metabolic Pathways	Down-Regulated Metabolic Pathways	Differential Secondary Metabolites [%]
Alcalde1/ Dongzao (fresh/fresh)	475	246	229	Stilbenoid, Flavone and Flavonol, Glutathione, Sulfur	Arginine, Linoleic acid, Alanine/ Aspartate /Glutamate, Proline, Porphyrin, Nitrogen, Cyanoamino acid, Valine/Leucine /Isoleucine, D-amino acid, Pyrimidine, Glyoxylate and decarboxylate, Butanoate, Propanoate	41.4
Alcalde1/Jinsi (fresh/MP)	375	192	183	Arginine and Proline, Purine alkaloids, Phenylpropanoid, Pyrimidine, Nucleotides, Tropane/piperidine/pyridine alkaloids, Betalain, Fatty acid, alpha-linoleic acid, glutathione, Purine, Stilbenoid	Sphingolipid, Flavonoid	40.5
Alcalde1/KFC (fresh/MP)	483	365	118	Flavonoid, Flavone and Flavonol, Phenylpropanoid, Stilbenoid	Tryptophan	53.7
Alcalde1/Lang (fresh/dry)	330	212	118	Flavonoid, Phenylpropanoid, Tryptophan, Phenylalanine, Glutathione	None	38.5
Alcalde1/Li (fresh/fresh)	336	206	130	Flavonoid, Sphingolipid, Quinone	Tyrosine, Isoquinoline alkaloids, Nucleotide	42.6
Alcalde1/Maya (fresh/fresh)	489	368	121	Phenylalanine, Phenylpropanoid, Flavonoid, Arginine and Proline	Flavone and Flavonol	42.5
Dongzao/JKW (fresh/MP)	564	316	248	Linoleic acid, Nucleotide, Purine, Arginine and Proline, Phenylalanine, D-amino acid,	Flavone and Flavonol	39.4
Dongzao/KFC (fresh/MP)	531	403	128	Arginine and Proline, Flavonoid, Isoquinoline alkaloid, Linoleic acid, Flavone and Flavonol, Tyrosine, Phenylalanine	Glutathione	42.4
Dongzao/Li (fresh/fresh)	487	260	227	Amino acid, Flavonoid, Linoleic acid, Sphingolipid	Phenylpropanoid, Flavone and Flavonol, Sulfur	50
Dongzao/Maya (fresh/fresh)	575	400	175	Linoleic acid, Quinone, Tyrosine, Arginine and proline, Tropane/piperidine/pyridine alkaloid, Isoquinoline alkaloid, Cyanoamino acid, Purine, Pyrimidine, Alanine/Aspartate/Glutamate, Nucleotide	None	42.4
JKW/Li (MP/fresh)	452	207	245	Flavonoid, Flavone and Flavonol	Linoleic acid	44.3
JKW/Maya (MP/fresh)	561	392	169	Phenylalanine, Sphingolipid, Tyrosine, Tryptophan, Arginine and proline	Purine, Phenylpropanoid	40
JKW/ZCW (MP/fresh)	384	160	224	Sphingolipid, Flavonoid, Flavone and Flavonol	Phenylalanine, Purine	37
Jinsi/JKW (dry/MP)	47	36	11		Amino acids, Tropane/piperidine/pyridine alkaloid, Glucosinolate, 2-oxocarboxylic acid	62.5
Jinsi/KFC (MP/MP)	494	379	115	Sphingolipid, Flavonoid, Flavone and Flavonol	Cyanoamino acid, Thiamine, Amino acids, Nucleotide, Pantothenate and CoA, Purine	46.8
Jinsi/Lang (MP/dry)	420	270	150	Fructose and Mannose, Nucleotide	Linoleic acid, Phenylpropanoid, D-amino acid	43.5
Jinsi/Maya (MP/fresh)	533	394	139	Phenylalanine, Tyrosine, Arginine and proline	Purine, Nucleotide, Glutathione	39.2
Jixin/KFC (dry/MP)	531	377	154	Flavone and Flavonol, Flavonoid, Phenylpropanoid, Tropane/piperidine /pyridine alkaloid, Phenylalanine	Tryptophan, Purine	56.7
KFC/Lang (MP/dry)	498	178	320	Glutathione	Phenylpropanoid, Flavone and Flavonol, Tyrosine	45.5
KFC/Li (MP/fresh)	492	154	338	Glutathione, D-amino acid	Phenylpropanoid, Flavonoid, Tropane/piperidine/pyridine alkaloid, Flavone and Flavonol	50.8
KFC/Maya (MP/fresh)	474	222	252	Linoleic acid, Tryptophan	Purine, Nucleotide, Flavonoid, Flavone and Flavonol	52.2
Lang/Li (dry/fresh)	355	149	206	Purine, Nucleotide	Flavonoid, Isoquinoline alkaloid, Phenylpropanoid, Tyrosine, Phenylalanine	50
Lang/Maya (dry/fresh)	437	265	172	Tyrosine, Phenylpropanoid, Linoleic acid, Arginine and proline	Phenylalanine, Glutathione, Purine, Flavonoid, Nucleotide	41.8
Lang/ZCW (dry/fresh)	445	181	264	Arginine and proline, Tyrosine, Glutathione, Tropane/piperidine/pyridine alkaloid, phenylpropanoid	Nucleotide, Purine, D-amino acid, Flavonoid	42.2
Li/Maya (fresh/fresh)	486	337	149	Linoleic acid, Tryptophan, Phenylpropanoid	Purine, Glutathione, Nucleotide	44.6
Li/Shanxi-Li (fresh/fresh)	42	36	6	None	None	75
Li/ZCW (fresh/fresh)	436	198	238	None	Glucosinolate, Amino acids, 2-Oxocarboxylic acid	42
Maya/ZCW (fresh/fresh)	519	130	389	Tropane/piperidine/pyridine alkaloids	Phenylalanine, Tryptophan, Arginine and proline, Flavonoid	53.1

**Table 2 plants-12-02313-t002:** Sample numbers of 11 jujube cultivars at different sites for widely targeted metabolomics in fall of 2022. Jinsi 2 and Jinsi 3 were considered as one cultivar (Jinsi, JS). Sandia and Hebei Dong were considered as one cultivar (Dongzao).

Cultivar	Code	Leyendecker (LK)	Los Lunas (LL)	Alcalde (AL)
Alcalde 1 (QYX)	Alc	1	2	2
Sandia/Hebei Dong	Dong	2	2	2
Jinsi 2	JS		2	2
Jinsi 3	JS	2		
Jixin	JX	2	2	2
Jinkuiwang	JKW	2	2	2
Kongfucui	KFC	2	2	2
Lang	Lang	2	2	1
Li	Li	1	2	2
Maya	Maya	2	2	2
Shanxi Li	SLi	2	2	2
Zaocuiwang	ZCW	2	2	1
Total		20	22	20

## Data Availability

The Jujube metabolomic data for peptide chemotyping is deposited to GNPS-MassIVE (Accession MSV000092019). All other data supporting the results of this study are presented in the manuscript, and supporting information is available from the corresponding author upon reasonable request.

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
