# Peer review of "Jujube Fruit Metabolomic Profiles Reveal Cultivar Differences and Function as Cultivar Fingerprints"

_plants, 2023, doi:10.3390/plants12122313_

Round 1
Reviewer 1 Report
The Authors should consider also the following articles.
1. Comparison of the sedative and hypnotic effects of flavonoids, saponins, and polysaccharides extracted from Semen Ziziphus jujube
Jian-Guo Jiang , Xiao-Juan Huang , Jian Chen & Qing-Sheng Lin
Natural Product Research, Volume 21, 2007 - Issue 4
2. Distribution of free amino acids, polyphenols and sugars of Ziziphus jujuba pulps harvested from plants grown in Tunisia
M. Elaloui, A. Laamouri, J. Fabre, C. Mathieu, G. Vilarem & B. Hasnaoui
Natural Product Research, Volume 29, 2015 - Issue 1
3. Investigation of the antimicrobial activities of Snakin-Z, a new cationic peptide derived from Zizyphus jujuba fruits
Fatemeh Daneshmand, Hadi Zare-Zardini & Leila Ebrahimi
Natural Product Research, Volume 27, 2013 - Issue 24
4. Antioxidant characteristics and antibacterial activity of native woody species from Catamarca, Argentina
María Emilia Lorenzo, Carina Noelia Casero, Patricia Elizabeth Gómez, Adrián Federico Segovia, Lara Carolina Figueroa, Alejandro Quiroga, María Laura Werning, Daniel Alberto Wunderlin & María Verónica Baroni
Natural Product Research, Volume 36, 2022 - Issue 4
Author Response
Dear Reviewer,
Thank you for taking your precious time to review our manuscript. We have included 3 suggested jujube papers and added two more papers in the text.
Thank you!
Reviewer 2 Report
In this manuscript, the authors presented the 11 cultivars of jujube fruits metabolomic profiles. The manuscript is well written, and the results are clearly presented. I do not have any substantial amendments to suggest. The manuscript has a merit to be published in Plants. To make this manuscript even better, please consider the following minor comments.
1. In Figures, 2, 3, 4, and 5, please use larger fonts to facilitate a better experience for the reviewers and readers.
2. Page 3, line 1; The authors should add more detailed information about amino acids, flavonoids, phenolics, alkaloids, terpenoids and nucleotides contained in jujube fruit. In particular, structural formulas of representative compounds among these should be added to the manuscript. For example, what are the typical flavonoids in jujube fruit? Also, what kind of compounds are typical phenols, alkaloids and terpenoids? The authors should consider the above points.
3. Page 12, line 4; The authors identified the cyclopeptide alkaloid sanjoinine A only from Jinsi and JKW cltivars using MS spectrum. However, it seems impossible to identify the stereochemistry only with the information obtained from fragmentation of MS. Also, it is not possible to identify it as sanjoinine A unless a comparison with a standard of sanjonine A is made. Did the authors perform a comparison with a standard of sanjoinine A? Authors should consider the above comments.
4. The authors should discuss why cyclopeptide alkaloids were identified only from Jinsi and JKW cltivars, and why they are biosynthesized.
Author Response
Dear Reviewer,
We are very grateful that you took your precious time and assessed our manuscript. Your comments and suggestions will definitely improve the quality of this manuscript. we have addressed all your comments and suggestions and the revision details are included in the file attached.
Thank you!
Shengrui

Round 2
Reviewer 1 Report
Manuscript can be accepted